# Influence of Grafting on Rootstock Rhizosphere Microbiome Assembly in *Rosa* sp. ‘Natal Brier’

**DOI:** 10.3390/biology12050663

**Published:** 2023-04-27

**Authors:** Dario X. Ramirez-Villacis, Pablo Erazo-Garcia, Juan Quijia-Pillajo, Sol Llerena-Llerena, Noelia Barriga-Medina, Corbin D. Jones, Antonio Leon-Reyes

**Affiliations:** 1Laboratorio de Biotecnología Agrícola y de Alimentos-Ingeniería en Agronomía, Universidad San Francisco de Quito USFQ, Quito 170109, Ecuador; 2Department of Biology, University of North Carolina, Chapel Hill, NC 27599-3280, USA; 3Department of Horticulture and Crop Science, The Ohio State University, Wooster, OH 43210, USA

**Keywords:** rose, *Rosa* spp., microbiome, rhizosphere, grafting, rootstock, scion, Natal Brier, Explorer^TM^, Freedom^TM^

## Abstract

**Simple Summary:**

Microorganisms are tightly associated with crops and can be pathogens or beneficials. Rose is the main ornamental crop worldwide. During production, rose varieties (over 100 in a single farm) are permanently grafted in a unique genotype as rootstock to improve plant performance. This work aimed to study the effect of grafting on root-associated microorganisms using next-generation DNA sequencing analysis. To this end, we have demonstrated that root-associated microorganisms of the rootstock (bacteria and fungi) will depend on the grafted genotype. In other words, a change in the variety will drive unique changes in the assembly of microorganisms at the root. This manuscript discusses differences in the bacterial and fungal communities when grafted and non-grafted and their potential impact on plant performance and agriculture.

**Abstract:**

The root microbiome is vital in plant development and health and is highly influenced by crop cultural practices. Rose (*Rosa* sp.) is the most popular cut flower worldwide. Grafting in rose production is a standard practice to increase yield, improve flower quality, or reduce root-associated pests and diseases. ‘Natal Brier’ is a standard rootstock used in most commercial operations in Ecuador and Colombia, leading countries in producing and exporting ornamentals. It is known that the rose scion genotype affects root biomass and the root exudate profile of grafted plants. However, little is known about the influence of the rose scion genotype on the rhizosphere microbiome. We examined the influence of grafting and scion genotype on the rhizosphere microbiome of the rootstock ‘Natal Brier’. The microbiomes of the non-grafted rootstock and the rootstock grafted with two red rose cultivars were assessed using 16S rRNA and ITS sequencing. Grafting changed microbial community structure and function. Further, analysis of grafted plant samples revealed that the scion genotype highly influences the rootstock microbiome. Under the presented experimental conditions, the rootstock ‘Natal Brier’ core microbiome consisted of 16 bacterial and 40 fungal taxa. Our results highlight that the scion genotype influences root microbe’s recruitment, which might also influence the functionality of assembled microbiomes.

## 1. Introduction

Rose (*Rosa* sp.) is a woody shrub that belongs to the Rosaceae family and is the most economically important cut flower crop worldwide. In South America, cut-rose production is mainly located in Ecuador and Colombia. In 2019, the export value of Ecuadorian roses was nearly 800 million dollars, and their principal destinies were the USA, Russia, and the Netherlands [1]. Traditionally, Ecuadorian farms grow from 50 to 120 different rose cultivars in a single farm. Red cultivars usually occupy about 30% to 50% of farm production land. Red roses are the most popular, and the cultivars Freedom^TM^ by Rosen Tantau and Explorer^TM^ by Interplant are the most popular red cultivars in Ecuadorian farms [2,3,4]. Grafting is a standard practice throughout commercial operations to increase yield, improve flower quality, or reduce root-associated diseases [5,6,7,8]. ‘Natal Brier’ (a hybrid of R. × *multiflora* and R. × *damascena*) has been the most used rootstock since 1990 [9,10].

Grafting is an ancient asexual propagation technique combining two plant genotypes to form a single composite plant. The composite plant has two sections, the scion (above-ground portion) and the rootstock (below-ground portion); each section has its genotype [11]. This technology widely propagates species that cannot be propagated by seed or cuttings. It also can improve growth, yield, and promote biotic/abiotic stress resistance [11]. Grafting has been reported to modify the metabolic profile of root exudates [12,13]. Root exudates play critical roles in plant nutrition, protection against pathogens, and microbiome assembly [14].

Thousands of microorganisms inhabit plants. The plant-associated microbial communities, their genomic material, and their interactions are known as the plant microbiome. Notably, the rhizosphere (soil zone closest to the root) is a hotspot for microbial activity influenced by root exudates [14]. The rhizosphere microbiome significantly modulates plant development and health [15] and can be influenced by phenological stage [16,17], biotic and abiotic stresses [18], or agriculture management practices [19,20]. A better understanding of microbiome dynamics in response to agricultural inputs and practices is vital to promoting the establishment of microbiomes that positively affect plant performance.

Few studies have looked at the effect of grafting on rhizosphere microbiome assembly. In *Vitis vinifera*, five rootstock genotypes grafted with the same scion genotype assemble different rhizosphere microbiomes; however, the observed rootstock genotype effect was farm dependent [21]. In *Vitis vinifera* cv. Lambrusco, potassium deficiency was associated with a rootstock genotype that assembles a microbiome depleted of potassium-solubilizing microorganisms [22]. Another study in grapevines showed that the scion cultivar highly influences the α-diversity and bacterial community structure at the rootstock rhizosphere and that the observed scion cultivar effect depends on the rootstock genotype evaluated [23]. Similarly, the apple scion genotype was reported to regulate root carbohydrate content and bacterial community structure and function [24]. In the case of herbaceous crops, it was reported that the rootstock genotype does not influence the leaf endophyte community assembly of tomato cv. ‘Momotaro-Haruka’ [25]. However, the structure of the rhizosphere microbiome of bottle gourd [*Lagenaria siceraria* (Molina) Standl.] was highly differentiated when grafted with a watermelon [*Citrullus lanatus* (Thunb.) Mansf.] scion. Interestingly, the rhizosphere microbiome shifted towards a structure similar to a non-grafted watermelon [26]. In the case of *Rosa* sp., prior studies have looked at endophyte communities across different genotypes [27] and root microbiota associated with rose replanting disease [28,29,30]. However, little is known about the influence of rose scion genotypes on the rootstock rhizosphere microbiome.

Due to the economic importance of grafting and the rootstock ‘Natal Brier’ for the Ecuadorian and Colombian rose industry, we aimed to understand the microbial composition and diversity at the rhizosphere of the rootstock ‘Natal Brier’ with and without grafting. To accomplish this objective, rhizosphere samples were collected from the Natal Brier rootstock with no graft and the Natal Brier rootstock grafted with two red rose cultivars (Explorer and Freedom). The 16S rRNA and ITS amplicon sequencing technology assessed bacterial and fungal community composition.

## 2. Materials and Method

### 2.1. Sample Collection

In May 2020, the sampling was conducted in Arbusta 2 (latitude N 0°2′53.66″; longitude O 78°11′30.85″), a farm in Tabacundo, Ecuador (Figure 1a,b). A total of 12 bulk soil (100 gr each) and 12 root samples were collected. Four soil samples and four root samples were collected from two plant combinations, Natal Brier (NB)-Freedom and NB-Explorer, and the non-grafted rootstock (NB) (Figure 1). Explorer and Freedom are red bud color cultivars (Figure 1d,e). Sampled plants were randomly selected. Bulk soil samples were collected between the plants and 20 to 30 cm deep. For all soil samples, soil texture was classified as sandy loam with an average pH of 5.8 ± 0.3, 5 ± 0.2% of organic matter, and electrical conductivity of 2 mS ± 0.2. The roots were removed from the plant (rhizosphere samples) with a shovel at 10 to 15 cm depth, and the roots were gently shaken to remove loose soil. All soil samples were placed in polyethylene bags and stored in a portable cooler until transported to the laboratory.

The roots were placed in Falcon tubes filled with phosphate buffer solution (PBS) for rhizosphere samples and vortexed for 10 min to detach the rhizosphere area. Roots were removed and tubes centrifuged at 500 rpm for 10 min. The supernatant was discarded, and the samples were stored at −20 °C until further use [23].

### 2.2. DNA Extraction, Library Preparation, and Sequencing

Genomic DNA was extracted using the DNeasy Powersoil^®^ Isolation Kit following the manufacturer’s instructions (Qiagen, Hilden, Germany). The DNA quality and concentration were assessed using a Qubit 4 (Invitrogen, Thermo Fisher Scientific, Waltham, MA, USA) and agarose gel electrophoresis.

The V3–V4 region of the 16S rRNA gene was amplified using the primers 338F (5′-ACTCCTACGGGAGGCAGCA-3′) and 806R (5′-GGACTACHVGGGTWTCTAAT-3′). Two barcodes and six frameshifts were added at the 5′ end of the primers [31]. Samples were run in triplicate, each reaction adding a unique mix of three frameshift primers to each plate. Each reaction contained 5 μL of KAPA Enhancers, 5 μL of KAPA Buffer A, 1.25 μL of each primer (5 μM), 0.375 μL of a mixture of peptide nucleic acids blocking rRNA genes, 0.5 μL of KAPA dNTPs, 0.2 μL of KAPA Robust Taq (KAPA Biosystems, Wilmington, MA, USA), 8 μL of dH2O, and 5 μL of DNA. The PCR protocol was: 95 °C for 60 s, 24 cycles at 95 °C for 15 s, 78 °C for 10 s, 50 °C for 30 s, 72 °C for 30 s, and finally, samples were held at 4 °C.

The ITS2 region was amplified using the primers ITS1-F (5′-CTTGGTCATTTAGAGGAAGTAA-3′) and ITS2 (5′-GCTGCGTTCTTCATCGATGC-3′). DNA samples were diluted to concentrations of 3.5 ng μL^−1^. Samples were run in triplicate. Each reaction contained 10 ng of DNA, 1 U of buffer, 0.3% bovine serum albumin, 2 mM of magnesium chloride, 200 μM of dNTPs, 300 nM of each primer, and 2 U of the DFS-Taq DNA polymerase (Bioron, Ludwigshafen, Germany). The PCR protocol was: 2 min at 94 °C, 25 cycles at 94 °C for 30 s, 55 °C for 30 s, 72 °C for 30 s, and 10 min at 72 °C to finish [31]. Then PCR products underwent enzymatic purification with exonucleases. A second triplicate amplification was performed with 3 μL of the purified PCR products and barcode-specific primers for the cycled samples under the previously mentioned conditions [31].

AMPure XP magnetic beads (Beckman Coulter, Brea, CA, USA) were used for bacterial and fungal amplicon purification and then quantified with the Qubit 2.0 fluorometer (Invitrogen, Carlsbad, CA, USA). Finally, the bacterial and fungal amplicons were bound in equal proportions and diluted to a concentration of 10 pM by sequencing, carried out on the Illumina MiSeq platform (Illumina, San Diego, CA, USA) using the 600-cycle V3 [31].

### 2.3. Data Analysis

16S rRNA and the ITS sequences were processed independently. MT-Toolbox software [32] was used to check sequence quality. Only sequence reads with 100% correct primer and a Q score over 20 were used. After the quality control, samples with less than 8000 16S rRNA reads or 17,000 ITS reads were discarded. The resulting sequences were clustered into amplicon sequence variants (ASVs) with the R package DADA2 version 1.8.1 [33]. The taxonomic assignment of each ASV was performed using the naïve Bayes *k*-mer method implemented in the DADA2 package using the Silva 132 database as a training reference. Fungal ITS forward sequence reads were processed using DADA2 [33] with default parameters. The taxonomic assignment of each ASV was performed using the naïve Bayes *k*-mer method implemented in the MOTHUR package [34] using the UNITE database [35] as a training reference.

### 2.4. Diversity Analysis, Community Composition, Community Function, and Statistical Analysis

16S rRNA and the ITS counts were rarified to 8000 reads per sample for bacteria and 17,000 reads per sample for fungi before further analysis with the ‘ohchibi’ package [36]. Alpha diversity indexes (Richness, Chao1, Shannon, Inverse Simpson, and Evenness) were calculated with the ‘diversity’ function from the vegan package version 2.5–3 [37]. Differences in the alpha diversity indexes between groups were analyzed with the Student’s *t*-test (*p* value of <0.05) using R base functions [38].

Beta diversity was visualized through a principal coordinate analysis (PCoA) constructed using the Bray–Curtis dissimilarity index. Permutational analysis of variance (PERMANOVA) was performed using the adonis function of the vegan package version 2.5–3 [37].

The relative abundance of bacterial phyla and fungal classes were exemplified using the stacked bar representation encoded in the ‘chibi.phylogram’ function of the ohchibi package [36].

To compare the enrichment of specific ASV in non-grafted versus grafted rhizosphere communities, we used the ‘DESeq2′ package [39] to run the model *abundance ∼ graft_status* using the raw count table. An ASV was considered statistically significant if it had a false discovery rate (FDR)-adjusted *p*-value of <0.01.

The functionality of bacterial and fungal communities was analyzed using the ‘Functional Annotation of Prokaryotic Taxa (FAPROTAX)’ [40] and the ‘FUNGuild’ tools [41], respectively. These tools used the taxonomic assignation of each ASV and established its functionality based on specific databases. Significant differences between non-grafted and grafted were established with a Student’s *t*-test (*p*-value of <0.05) using R base functions [38].

An ASV was considered part of the core microbiome if it was present in all samples (100% occupancy) with at least 5 reads. The sequence for the core ASVs were used to construct a maximum likelihood phylogenetic tree with the online tool ‘RAxML BlackBox’ [42]. We then used the Interactive Tree of Life (iTOL) web-based tool (https://itol.embl.de/) (accessed on 2 February 2021), to visualize the tree.

## 3. Results

### 3.1. Alpha Diversity and Taxonomic Microbial Composition of Bulk Soil and the Rhizosphere of the Rootstock ‘Natal Brier’

16S rRNA and ITS amplicon sequencing were performed on rhizosphere and bulk soil samples from the two plant combinations, Natal Brier (NB)-Freedom and NB-Explorer, and the non-grated rootstock (NB). 16S rRNA amplicon sequencing rendered 2107, 1514, and 1924 bacterial ASVs in NB, NB-Freedom, and NB-Explorer, respectively. Only 1588, 891, and 1302 bacterial ASVs were unique for NB, NB-Freedom, and NB-Explorer, respectively. ITS amplicon sequencing produced 676, 934, and 1028 fungal ASVs in NB, NB-Explorer, and NB-Freedom, respectively. Only 288, 483, and 542 fungal ASVs were unique for NB, NB-Explorer, and NB-Freedom, respectively. A total of 202 bacterial ASVs and 223 fungal ASVs were shared among all three rootstock systems (NB-Freedom, NB-Explorer, and NB).

Bulk soil and rhizosphere samples were separated using a principal component analysis (PCo2) (Figure 2). The rhizosphere and bulk soil assembled different bacterial communities (Figure 2a). In contrast, the fungal communities in the rhizosphere and bulk soil samples clustered similarly (Figure 2c). There was no difference between the bacterial α-diversity of bulk soil and the rhizosphere (Table 1). However, we observed that bulk soil has higher fungal α-diversity than the rhizosphere (*p*-value < 0.05) (Table 1). In soil and rhizosphere samples, the bacterial communities were dominated by the phyla Actinobacteria, Bacteroidetes, and Proteobacteria (Figure 2b), and the fungal communities were dominated by Ascomycota (Appendix A) at the class level by Sordariomycetes, Dothideomycetes, and Mortierellomycetes (Figure 2d).

### 3.2. Alpha Diversity, Taxonomic Composition, and Function of Microbial Communities Associated with the Grafted and Non-Grafted ‘Natal Brier’ Rootstock

The influence of the graft scion on microbial communities was evaluated in the rhizosphere samples. The rhizosphere bacterial and fungal communities in the non-grafted rootstock differed from those in the grafted rootstock. (Figure 3a,c). Bacterial α-diversity indexes were not statistically different but tended to be lower in the grafted rootstock (Table 2). In contrast, fungal α-diversity was higher in the grafted rootstock (*p*-value < 0.05) (Table 2). Ten bacterial phyla and twelve fungal classes dominated the rhizosphere microbiome of grafted and non-grafted plants. Proteobacteria was the most abundant bacterial phylum, followed by Bacteroidetes, Actinobacteria, and Acidobacteria. Sordariomycetes were the most abundant fungal class (~50%) (Figure 3b,d).

Differential abundance analysis was performed to identify microbial taxa significantly enriched in grafted and non-grafted plants (Figure 4a,b). The rhizospheres of non-grafted plants enriched seven bacterial ASVs assigned to the phyla Actinobacteria (ASV446), Alphaproteobacteria (ASV40), Bacteroidetes (ASV20 and ASV417), Gemmatimonadetes (ASV509), and Verrucomicrobia (ASV639 and ASV278) (Figure 4a) and seven fungal ASVs classified as *Alternaria* spp. (ASV26), *Blastobotrys* spp. (ASV171), *Chaetomiaceae* spp. (ASV284), *Cladosporium* spp., *Disculoides* spp. (ASV309), *Fusarium* spp. (ASV189), and *Plectosphaerella* spp. (ASV386) (Figure 4b). The bacterial ASV40 was classified as *Sphingomonadaceae* spp. and was the most enriched taxa in non-grafted plants. *Chaetomiaceae* spp., *Disculoides* spp., and *Fusarium* spp. were the most enriched fungal taxa in non-grafted plants. Conversely, the rhizospheres of grafted plants enriched 25 bacterial ASVs classified into ten phyla and 35 fungal ASVs classified into 23 genera (Figure 4a,b).

Differentially abundant bacterial and fungal taxa (ASVs) were assigned into functional categories using the FAPROTAX database [40] and the FUNGuild database [41], respectively. The abundance of bacteria involved in chemo heterotrophy, aerobic chemo heterotrophy, cellulolysis, and nitrate reduction was higher in non-grafted plants. On the other hand, the abundance of bacteria involved in nitrification, aerobic nitrite oxidation, intracellular parasites, and non-photosynthetic cyanobacteria was higher in grafted plants (Figure 5a). The abundance of fungal taxa classified as plant pathogens was higher in the non-grafted plants. In contrast, the fungal taxa in grafted plants were enriched into ectomycorrhizal and arbuscular mycorrhizal categories (Figure 5b).

### 3.3. Microbial Community Composition in the Rhizospheres of ‘Natal Brier’ Rootstock Grafted with Cultivars ‘Freedom’ and ‘Explorer’ and Their Core Microbiome

We observed an effect of the scion genotype on bacterial and fungal communities in the rootstock rhizosphere. Principal coordinate analysis showed that NB-Explorer rhizosphere samples separate from NB-Freedom samples along the first principal coordinate (PCo1) (Figure 6a,c). There were non-significant differences in bacterial and fungal α-diversity between plants grafted with Explorer and Freedom cultivars, but plants grafted with Explorer tended to have higher values on the α-diversity indexes evaluated (Table 3).

NB-Explorer and NB-Freedom plants shared 16 bacterial and 40 fungal taxa in all samples. These prevalent taxa could represent the “core microbiome” of the rose rootstock ‘Natal Brier’. The 16 bacterial taxa belong to the phyla Proteobacteria (8), Acidobacteria (1), Actinobacteria (4), Firmicutes (2), and Patescibacteria (1) (Figure 7a). The 40 fungal taxa belong to 11 classes: Sordariomycetes (25), Eurotiomycetes (3), Dothideomycetes (1), Ascomycota_unclassified (1), Orbiliomycetes (1), Tremellomycetes (2), Tremellomycetes (1), Archaeorhizomycetes (1), Mortierellomycetes (3), Malasseziomycetes (1), and unclassified (1) (Figure 7b). The number of unique taxa was higher in NB-Explorer (77 bacterial and 93 fungal taxa) than in NB-Freedom (15 bacterial and 51 fungal taxa).

## 4. Discussion

Rose grafting is a propagation technique widely used to improve plant performance and flower quality [7,8,43]. The superior performance of grafted rose plants depends on rootstock selection, growing media, geographic region, environmental conditions, and culturing practices [7,44,45,46]. The rootstock genotype selection influences the reblooming time, scion nutritional status, and dry matter partitioning [47,48]. Several reports studying rose scion–rootstock interaction are available; however, to our knowledge, none have focused on the rootstock microbiome and the influences of scion genotype. We described the rhizosphere microbiome of the non-grafted rootstock ‘Natal Brier’ and how it changes upon grafting with red cultivars ’Explorer’ and ‘Freedom.’

### 4.1. The Rhizosphere Microbiome of Rosa sp. ‘Natal Brier’

The plant and its associated microbiota are thought to function as an entire entity. The plant-associated microbiota serves the plant by providing additional processes that facilitate or improve plant development, health, and adaptation [49]. Understanding the effects of grafting on the rootstock microbiome is necessary to unveil the mechanism behind the superior performance of grafted plants. The bacterial and fungal communities of the rootstock’ Natal Brier’ are dominated by the phyla Proteobacteria and Sordariomycetes, respectively (Figure 3b,d). Similarly, the bacterial microbiome of *Rosa corymbifera* ‘Laxa’, a rootstock sensitive to rose replanting disease (RRD) used for garden rose propagation, is dominated by the phylum Proteobacteria. However, the soil source determines the dominant phyla of the fungal microbiome of *R. corymbifera* ‘Laxa’ [28]. Thus, the soil source is an important factor to consider for further understanding of rootstock microbiome assembly.

### 4.2. Grafting Changes the Rhizosphere Microbiome of Rose sp. ‘Natal Brier’

Root exudates play vital roles in plant nutrition, protection against pathogens, and microbiome assembly [14]. Upon soil colonization, root compartments are occupied by soilborne microorganisms. Primary and secondary metabolites regulate plant–microbe interactions at the root zone into the rhizosphere [15]. Shifts in the metabolic profile of root exudates are associated with plant developmental stage, health status, or stress responses [50]. In addition, the changes in root exudate composition are accompanied by changes in microbiome structure and diversity [17,51]. In *R. corymbifera* ‘Laxa’, the rhizosphere microbiome and root metabolite content change in response to rose replant disease (RRD) [28]. Grafting induces changes in microbiome structure, microbial activity, and root exudate composition of the rootstock [12,13,26,52]. Consistent with previous observations in other plant species, grafting also changed the rhizosphere microbiome composition of the rootstock ‘Natal Brier’ (Figure 3). Other reports have shown that bacterial *diversity* in the rhizospheres of grafted plants is higher than that of non-grafted rootstock rhizospheres [53]. However, we observed that grafting only increased the diversity of the fungal microbiome but not bacterial diversity (Table 2).

Bacterial functional prediction in grafted plants included nitrification, aerobic nitrite oxidation, intracellular parasites, and non-photosynthetic cyanobacteria. Accordingly, the genus Nitrosospira and Nitrospira were enriched. Nitrosospira converts ammonia (NH_3_) to nitrite (NO^2−^) and Nitrospira converts nitrite to nitrate (NO_3_) during the nitrification process [54]. The rhizospheres of grafted plants was enriched with nitrogen-fixing taxa (Frankiales and Rhizobiales) [55,56]. Other beneficial bacterial taxa included Chitinophagaceae and Saccharimonadales [57,58]. The increase in the abundance of nitrogen-fixing bacteria in grafted plants suggests a mechanism for superior performance in grafted plants. Likewise, bacterial taxa involved in nitrogen cycling can significantly impact nitrogen use efficiency.

Conversely, the functions assigned to bacteria enriched in the non-grafted rootstock were chemoheterotroph, aerobic chemoheterotroph, cellulolysis, and nitrate reduction. For example, *Jatrophihabitans soli* KIS75-12T was reported to hydrolyze urea into ammonia [59]. In addition, Gemmatimonadaceae abundance correlates with N fertilization [60]. Other bacterial taxa reported as beneficial are *Sphingomonadaceae* and *Verrucomicrobia* [61]. Species of genus *Niastella* have been isolated from soil, and *Niastella gongjuensis* sp. nov was reported to reduce nitrate [62].

The fungal taxa enriched in the rhizospheres of the non-grafted plants were assigned to the plant pathogen category. These taxa were classified as *Alternaria* spp., *Blastobotrys* spp., *Chaetomiaceae* spp., *Cladosporium* spp., *Disculoides* spp., *Fusarium* spp., and *Plectosphaerella* spp. Among them, only *Cladosporium* spp., *Alternaria* spp. and *Fusarium* spp have been previously reported to cause disease in roses. *Cladosporium cladosporioides* causes petal discoloration and necrosis [63], *Alternaria* spp. causes bud blight and leaf black spot [64], and *Fusarium oxysporum* causes dry rot diseases in rose seedlings [65].

In contrast, grafted plants’ taxa enriched in the rhizosphere were assigned to guilds ectomycorrhizal and arbuscular mycorrhizal. In roses, arbuscular mycorrhizal colonization stimulates nutrient and water uptake, biomass accumulation, and tolerance to environmental stress (drought and alkaline irrigation) and disease [66,67,68,69,70,71,72,73]. Among the enriched fungal taxa in grafted plants, some *Mortierella*, *Chaetomium*, *Humicola*, *Metarhizium*, *Schizothecium*, and *Volutella* species have been reported to benefit plant growth or health [74,75,76,77,78,79]. *Volutella citronella* is nematophagous, and *Metarhizium* spp. are entomopathogens [78,80]. Thelonectria and Gymnostellatospora species are mostly saprophytes [81,82]. Only *Thelonectria rubi* is a plant pathogen [81]. Penicillium and Acremonium were the only taxa reported as a pathogen for cut roses [63,83,84]. Acrostalagmus and Sarocladium were reported as plant pathogens in roses [85,86]. Thus, our findings suggest that grafted roses tend to recruit microbiomes with greater beneficial potential for plant growth and health.

### 4.3. Scion Genotype Determines the Changes in the Rhizosphere Microbiome of Rootstock Rose sp. ‘Natal Brier’

The performance of rose cultivars grown on their own roots is inferior to that of plants grafted onto a rootstock. The superior performance of grafted roses results from the additive effect of the scion and rootstock genotypes. Thus, the superiority of a grafted rose onto a specific rootstock depends on the scion vigor [8]. Other studies showed that the scion genotype could modify root biomass, transcriptome, and root exudate composition [87,88,89]. Likewise, the scion genotype can mediate microbiome recruitment [52]. To gain more insight into the scion genotype effect on the rhizosphere microbiome, we analyzed the microbiome of the red cultivars ‘Freedom’ and ‘Explorer’ grown onto the rootstock ‘Natal Brier’. Scion genotype determined the rhizosphere microbiome composition (Figure 6). Although no significant differences were found, both fungal and bacterial alpha diversity of the cultivar ‘Explorer’ tended to be higher than of the cultivar ‘Freedom’. The soil has been highlighted as a reservoir of microbial diversity [90], and the root exudates drive microbiome recruitment. As previously discussed, the scion genotype is an essential factor in the composition of the root exudates and rhizosphere microbiome. Accordingly, the rhizosphere microbiome of the grafted rootstock ‘Natal Brier’ showed taxa unique for each rootstock–scion combination, and the group of taxa consistently present is considered as a core microbiome (Figure 7). The number of unique microbial and fungal taxa was higher in the rootstock grafted with the cultivar ‘Explorer’. The bacterial core microbiome consists of taxa belonging to the phyla Proteobacteria, Acidobacteria, Actinobacteria, Firmicutes, and Patescibacteria. The fungal core microbiome consists of taxa belonging to the phyla Sordariomycetes, Eurotiomycetes, Dothideomycetes, Ascomycota_unclassified, Orbiliomycetes, Tremellomycetes, Tremellomycetes, Archaeorhizomycetes, Mortierellomycetes, and Malasseziomycetes. Further investigation is needed to reveal the potential contributions of the microbiome to the yield and vase life of these red cut-rose cultivars.

## 5. Conclusions

Grafting is a propagation technique that is widely used to improve productivity and flower quality in cut-rose production. The superior vigor of grafted rose cultivars results from the additive effect of the rootstock and scion. In grafted plants, the scion may mediate microbiome assembly in the rootstock [52]. Here, we described the effect of grafting on the rhizosphere microbiome of the rootstock ‘Natal Brier’. Our data suggest that the rhizosphere microbiome of the rootstock changes upon grafting. Grafting leads to the enrichment of plant-growth-promoting taxa in the rhizosphere. Changes in microbial community composition also modulated the microbiome functions. A wider range of microbiome functions potentially contributes to the superior performance of grafted plants. According to our data, the scion cultivar modulates the recruitment of microbiomes with beneficial effects. Thus, assessment of a wider diversity of scions will contribute to discovering microbiomes with superior beneficial functions. Understanding the potential functions of the plant microbiome opens an avenue to exploit the full potential of crops by leveraging the microbiome through adapting current breeding strategies, agronomic practices, and microbial products. The rose rootstock’ Natal Brier’ is a suitable biological model for basic and applied research to study the effects of grafting on the microbiome due to the vast number of scion genotypes available and its economic importance for countries cultivating cut roses [10].

## Figures and Tables

**Figure 1 biology-12-00663-f001:**
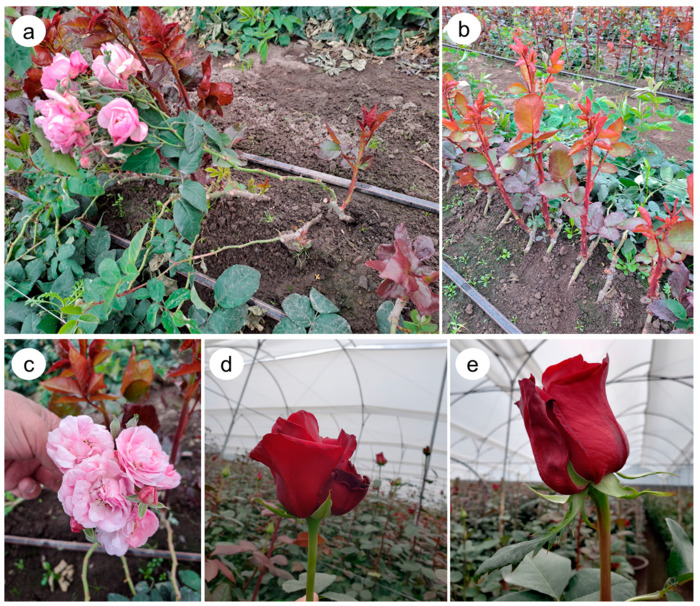
*Rosa* sp. ‘Explorer’ grafted onto ‘Natal Brier’ (NB) rootstock; the rootstock shoot is bent and blooming (**a**). Farm in Ecuador growing *Rosa* sp. ‘Freedom’ grafted onto ‘Natal Brier’ rootstock (**b**). Blooms produced by ‘Natal Brier’ (**c**), ‘NB-Explorer’ (**d**), and ‘NB-Freedom’ (**e**).

**Figure 2 biology-12-00663-f002:**
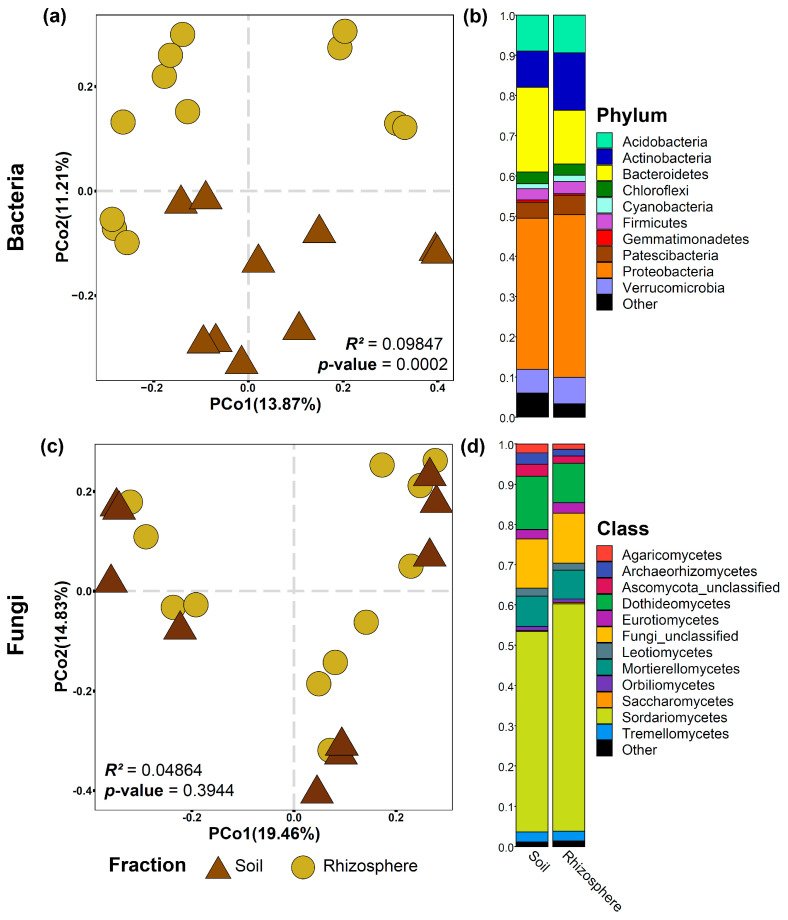
Beta diversity analysis and microbiome composition of bacterial (**a**,**b**) and fungal (**c**,**d**) communities for soil and rose rhizosphere samples. Beta diversity analysis was based on the Bray–Curtis dissimilarity matrix. The differentiation observed in bacteria is much more pronounced than in fungi.

**Figure 3 biology-12-00663-f003:**
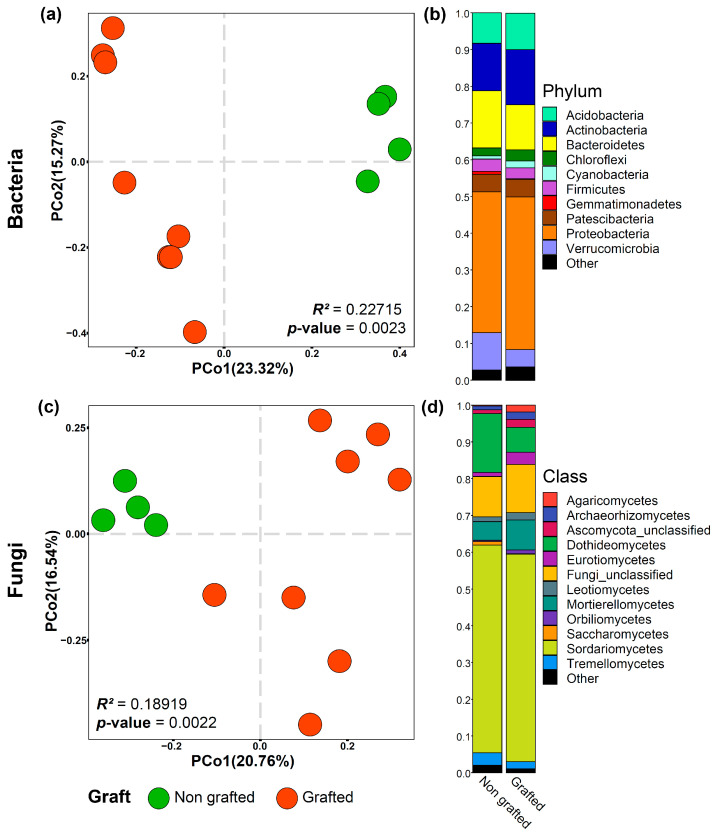
Effect of grafting on the rhizosphere microbiome composition of rootstock ‘Natal Brier’. Beta diversity analysis and microbiome composition of bacterial (**a**,**b**) and fungal (**c**,**d**) communities for grafted and non-grafted roses. Beta diversity analysis was based on the Bray–Curtis dissimilarity matrix. In both cases, the communities have clear differentiation due to grafting.

**Figure 4 biology-12-00663-f004:**
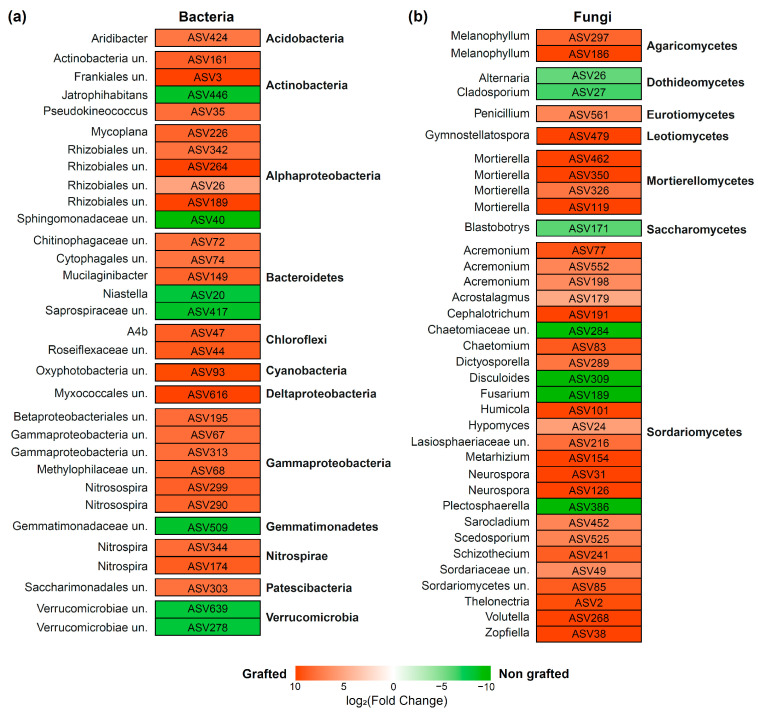
Differential abundance analysis of bacteria (**a**) and fungi (**b**) for grafted versus non-grafted roses.

**Figure 5 biology-12-00663-f005:**
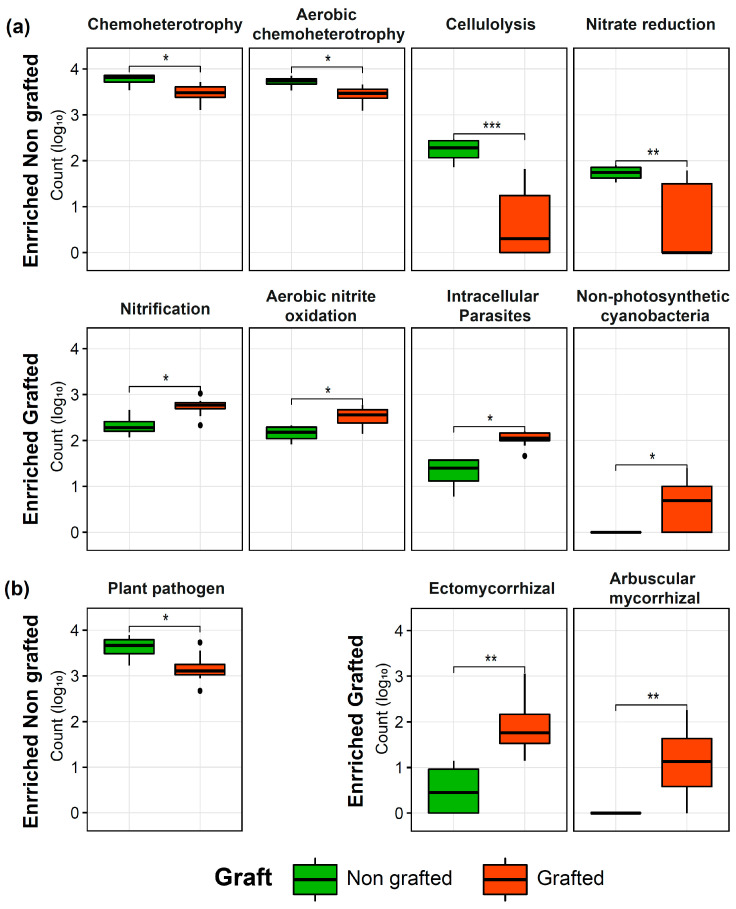
Effect of grafting on the functionality of the bacterial (**a**) and fungal (**b**) rhizosphere microbiome. Beneficial functional categories are enriched as a result of grafting. Significant differences were calculated by Student *t*-Test: * *p*-value < 0.05; ** *p*-value < 0.01; *** *p*-value < 0.001.

**Figure 6 biology-12-00663-f006:**
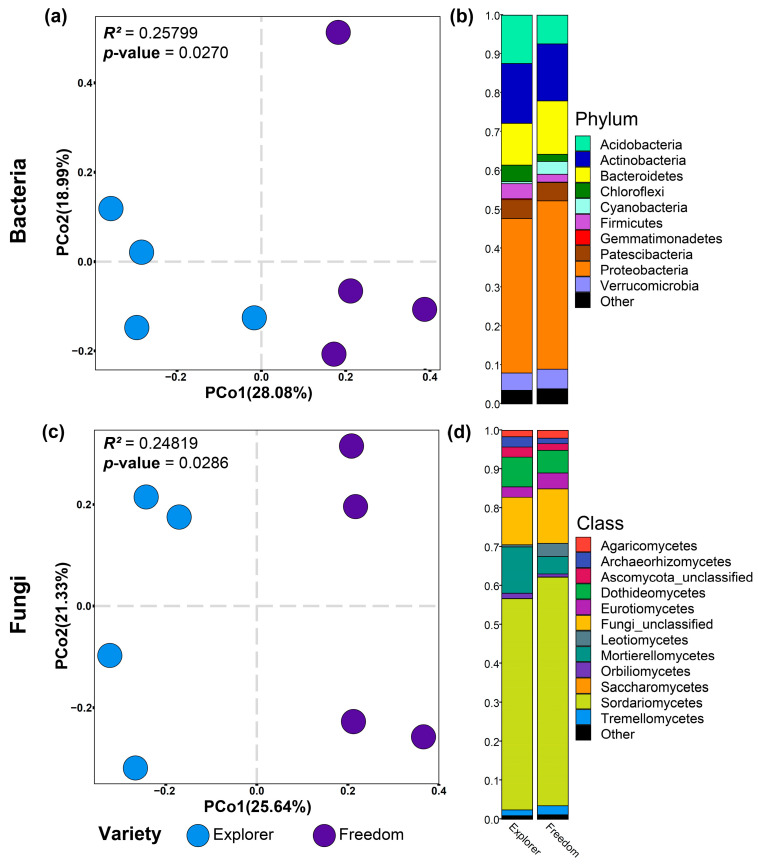
Effect of the scion cultivar on the rhizosphere microbiome of rootstock ‘Natal Brier’. Beta diversity analysis and microbiome composition of bacterial (**a**,**b**) and fungal (**c**,**d**) communities for Natal Brier (NB)-Freedom and NB-Explorer samples. Beta diversity analysis was based on the Bray–Curtis dissimilarity matrix. There was strong differentiation among the scion cultivars.

**Figure 7 biology-12-00663-f007:**
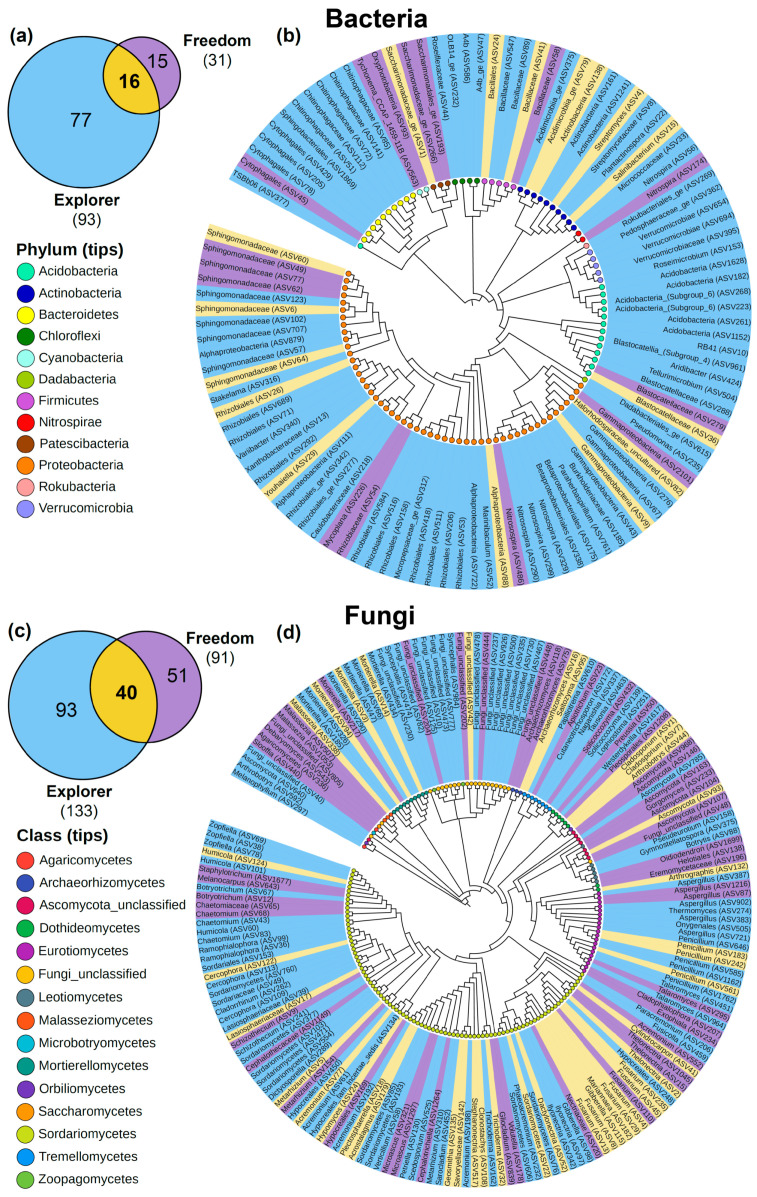
Bacterial (**a**,**b**) and fungal (**c**,**d**) core microbiomes for rhizospheres of two red rose cultivars. The core microbiome of the rootstock is highly influenced by the scion cultivar.

**Table 1 biology-12-00663-t001:** Alpha diversity indexes for soil and rhizosphere.

**Bacteria**					
**Fraction**	**Richness**	**Chao1**	**Shannon**	**Inverse Simpson**	**Evenness**
Soil	800.30 ± 310.99	833.29 ± 333.85	6.13 ± 0.56	308.23 ± 148.14	0.93 ± 0.01
Rhizosphere	672.67 ± 282.47	693.54 ± 298.81	5.93 ± 0.58	244.55 ± 139.39	0.93 ± 0.02
No significant differences found					
**Fungi**					
**Fraction**	**Richness ***	**Chao1 ***	**Shannon**	**Inverse Simpson**	**Evenness**
Soil	484.70 ± 95.58	532.63 ± 117.35	4.42 ± 0.47	35.05 ± 20.05	0.72 ± 0.06
Rhizosphere	379.00 ± 97.90	405.80 ± 112.84	4.24 ± 0.34	30.51 ± 12.12	0.72 ± 0.05

* Significant differences (*t*-Test, *p*-value < 0.05).

**Table 2 biology-12-00663-t002:** Alpha diversity indexes for rhizospheres of non-grafted and grafted plants.

**Bacteria**					
**Graft**	**Richness**	**Chao1**	**Shannon**	**Inverse Simpson**	**Evenness**
Non grafted	744.25 ± 304.84	767.73 ± 326.03	6.07 ± 0.57	284.25 ± 160.82	0.93 ± 0.02
Grafted	636.88 ± 284.9	656.45 ± 300.06	5.86 ± 0.61	224.71 ± 134.54	0.92 ± 0.02
No significant differences found					
**Fungi**					
**Graft**	**Richness ***	**Chao1 ***	**Shannon**	**Inverse Simpson**	**Evenness**
Non-grafted	288 ± 82.16	301.99 ± 95.81	4.16 ± 0.3	33.02 ± 11.88	0.74 ± 0.03
Grafted	424.5 ± 71.2	457.7 ± 82.7	4.27 ± 0.36	29.25 ± 12.85	0.71 ± 0.05

* Significant differences (*t*-Test, *p*-value < 0.05).

**Table 3 biology-12-00663-t003:** Alpha diversity indexes for rhizospheres of the two grafted varieties.

**Bacteria**					
**Variety**	**Richness**	**Chao1**	**Shannon**	**Inverse Simpson**	**Evenness**
Explorer	764.75 ± 296.75	793.98 ± 314.65	6.09 ± 0.46	256.68 ± 140.53	0.93 ± 0.02
Freedom	509.00 ± 240.26	518.92 ± 246.27	5.62 ± 0.71	192.73 ± 140.56	0.92 ± 0.03
No significant differences found					
**Fungi**					
**Variety**	**Richness**	**Chao1**	**Shannon**	**Inverse Simpson**	**Evenness**
Explorer	430.5 ± 44.67	463.99 ± 53.81	4.37 ± 0.29	32.53 ± 12.83	0.72 ± 0.04
Freedom	418.5 ± 98.68	451.41 ± 113.83	4.17 ± 0.45	25.97 ± 13.85	0.69 ± 0.07
No significant differences found					

## Data Availability

Amplicon sequencing data are available at the NCBI Sequence Read Archive (project PRJNA893836). Count tables and relevant data files can be found at GitHub https://github.com/darioxr/grafting_rose_microbiome (accessed on 10 December 2022).

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
