# Peer review of "Influence of Grafting on Rootstock Rhizosphere Microbiome Assembly in Rosa sp. ‘Natal Brier’"

_biology, 2023, doi:10.3390/biology12050663_

Round 1

Reviewer 1 Report

Comment No. 1: The entire manuscript is well written, materials and methods are provided in detail with appropriate references and the result and discussion part is well explained and discussed and bringing all necessary information.

Comment No. 2: The authors should maintain the citing references throughout the manuscript in a chronological order.

Author Response

Comment No. 1: The entire manuscript is well written, materials and methods are provided in detail with appropriate references and the result and discussion part is well explained and discussed and bringing all necessary information.

Comment No. 2: The authors should maintain the citing references throughout the manuscript in a chronological order.

Thanks for your comments, we have review again all references.

Reviewer 2 Report

Thank you for giving me the opportunity to review this work.

The work is good; however, they have several areas of opportunity for improvement. Here are my observations:

·         In the introduction section, specifically in lines 85-86, the exudates are mentioned, however, it is not pertinent because the exudates of the grafted and non-grafted plants were not evaluated; It would be worth putting a paragraph on what the literature says about it, but not in this part that is close to the objective of the study.

·         The M&M section is not very specific, if someone wanted to redo the experiment it would be impossible, due to missing details. In line 95 they should give the coordinates of the site where the soil was sampled. On line 96, they do not say how many grams of soil were collected from each sample. It is also important that they give details of the characterization of the soil, since depending on its physical and chemical characteristics is the prevalence of some bacteria and fungi.

·         In the conclusion of the work (line 419-422), you should not put something that was not done and that is only a conjecture (because they did not put a paragraph that talks about works that evidence the issue of exudates). Please redo the conclusion.

Author Response

Thank you for giving me the opportunity to review this work.

The work is good; however, they have several areas of opportunity for improvement. Here are my observations:

  • In the introduction section, specifically in lines 85-86, the exudates are mentioned, however, it is not pertinent because the exudates of the grafted and non-grafted plants were not evaluated; It would be worth putting a paragraph on what the literature says about it, but not in this part that is close to the objective of the study.

Thanks for your comments, you are correct we have eliminated the sentences were exudates are mentioned.

  • The M&M section is not very specific, if someone wanted to redo the experiment it would be impossible, due to missing details. In line 95 they should give the coordinates of the site where the soil was sampled. On line 96, they do not say how many grams of soil were collected from each sample. It is also important that they give details of the characterization of the soil, since depending on its physical and chemical characteristics is the prevalence of some bacteria and fungi.

Thanks. We have improved the details in the materials and methods and given a general overview of soil characteristics.

  • In the conclusion of the work (line 419-422), you should not put something that was not done and that is only a conjecture (because they did not put a paragraph that talks about works that evidence the issue of exudates). Please redo the conclusion.

Thanks. We have deleted that sentences regarding the exudates.

Reviewer 3 Report

The manuscript contains interesting research results. The manuscript should be adapted to the requirements of the journal. For example, citations are not correctly given.

Comments

Line 95, there is no information on what year the study was conducted and where.

Line 183, please provide full details of the manufacturer of the statistical software used to compile the data.

References, please remove publications older than 10 years.

Author Response

The manuscript contains interesting research results. The manuscript should be adapted to the requirements of the journal. For example, citations are not correctly given.

Comments

Line 95, there is no information on what year the study was conducted and where.

Thanks, we have included that in the materials and methods

Line 183, please provide full details of the manufacturer of the statistical software used to compile the data.

Added with track changes

References, please remove publications older than 10 years

Thanks , but we consider that some manuscripts included are relevant in this area and are older than 10 years old.

Reviewer 4 Report

The  present article  entitled "Influence of grafting on rootstock rhizosphere microbiome assembly in Rosa sp. 'Natal Brier'"  cover a very  interesting theme. Although article is well  written and clearly presented,

However  their is some quarries which need to be clarified

-Fig.2 and Fig 5  incase of bacteria , authors have considered phyla, while class  for fungal group. So please make the uniformity. Author should present here the genera  and  mentioned in text about the phyla.

-Fig .3- I encourage author to present the figure in bar form. 

- -Arrange the refrences in the journal format 

Author Response

dThe  present article  entitled "Influence of grafting on rootstock rhizosphere microbiome assembly in Rosa sp. 'Natal Brier'"  cover a very  interesting theme. Although article is well  written and clearly presented,

However  their is some quarries which need to be clarified

-Fig.2 and Fig 5  incase of bacteria , authors have considered phyla, while class  for fungal group. So please make the uniformity. Author should present here the genera  and  mentioned in text about the phyla.

Thanks for the comment. To facilitate the comparison, we add a supplementary figure (S1) with the phyla composition for fungi for the three comparisons: compartment (Fig 1), grafting (Fig 2), and variety (Fig 5). However, at the phyla level is difficult to pinpoint changes since two phyla (Ascomycota y Basidiomycota) dominate all samples in general (~80% of relative abundance), so to have better contrast, we present at class level what makes it easier to compare. This strategy has been seen in other papers on microbiome analysis (Finkel et al., 2019; Ramirez et al., 2023).

Also, a genus-level comparison is more difficult since there are 494 fungi genera in the data set, which makes it challenging to visualize the composition in a stack bar plot. Therefore, to facilitate the analysis at lower taxonomic levels, we first identified significant ASV and annotated them at the genus level for Fig 3 and Fig 6.

Finkel, O. M., Salas-González, I., Castrillo, G., Spaepen, S., Law, T. F., Teixeira, P. J. P. L., ... & Dangl, J. L. (2019). The effects of soil phosphorus content on plant microbiota are driven by the plant phosphate starvation response. PLoS Biology, 17(11), e3000534.

Ramirez-Villacis, D. X., Pinos-Leon, A., Vega-Polo, P., Salas-González, I., Jones, C. D., & Torres, M. D. L. (2023). Untangling the Effects of Plant Genotype and Soil Conditions on the Assembly of Bacterial and Fungal Communities in the Rhizosphere of the Wild Andean Blueberry (Vaccinium floribundum Kunth). Microorganisms, 11(2), 399.

-Fig .3- I encourage author to present the figure in bar form. 

Thanks for the suggestion. Using a heatmap is a common way to present differential abundance analysis, and we think the figure is clear in the present format.

- -Arrange the refrences in the journal format

Done.